# Adhesion Deregulation in Acute Myeloid Leukaemia

**DOI:** 10.3390/cells8010066

**Published:** 2019-01-17

**Authors:** Alicja M. Gruszka, Debora Valli, Cecilia Restelli, Myriam Alcalay

**Affiliations:** 1Department of Experimental Oncology, Istituto Europeo di Oncologia IRCCS, Via Adamello 16, 20 139 Milan, Italy; debora.valli@ieo.it (D.V.); cecilia.restelli@ieo.it (C.R.); myriam.alcalay@ieo.it (M.A.); 2Department of Oncology and Hemato-Oncology, University of Milan, Via Festa del Perdono 7, 20 122 Milan, Italy

**Keywords:** acute myeloid leukaemia, adhesion molecules, EMT

## Abstract

Cell adhesion is a process through which cells interact with and attach to neighboring cells or matrix using specialized surface cell adhesion molecules (AMs). Adhesion plays an important role in normal haematopoiesis and in acute myeloid leukaemia (AML). AML blasts express many of the AMs identified on normal haematopoietic precursors. Differential expression of AMs between normal haematopoietic cells and leukaemic blasts has been documented to a variable extent, likely reflecting the heterogeneity of the disease. AMs govern a variety of processes within the bone marrow (BM), such as migration, homing, and quiescence. AML blasts home to the BM, as the AM-mediated interaction with the niche protects them from chemotherapeutic agents. On the contrary, they detach from the niches and move from the BM into the peripheral blood to colonize other sites, i.e., the spleen and liver, possibly in a process that is reminiscent of epithelial-to-mesenchymal-transition in metastatic solid cancers. The expression of AMs has a prognostic impact and there are ongoing efforts to therapeutically target adhesion in the fight against leukaemia.

## 1. Introduction

In multicellular organisms, cells adhere to each other to form tissues, organs, and systems. For such a high degree of organization, it is essential to establish junctions between cells as well as between cells and extracellular matrix (ECM). Cell-to-cell and cell-to-matrix adhesions are the result of the interaction between ECM components (such as collagen, glycosaminoglycans, proteoglycans, fibronectin, and laminin), membrane-associated adhesion molecules (AMs) and the anchored cytoskeleton. These connections not only maintain a strict tissue structure, but also play a role in cell migration, differentiation, and communication. In normal tissues, AMs expression is tightly regulated. However, aberrant expression of AMs occurs during disease and in malignant transformation.

Any of numerous specialized trans-membrane molecules on the cell surface that bring about adhesion by binding to molecules on the surface of other cells or to ECM are defined as AMs. There are four main families of AMs: integrins, selectins, cadherins, and the superfamily of immunoglobulins (IgSF). Integrins are heterodimeric trans-membrane glycoproteins consisting of an α and a β subunit, assembled in different ways to generate a wide range of dimers (reviewed in [1]). Each type of integrin has a unique repertoire of ligands that can be either ECM molecules or trans-membrane cell AMs [2,3,4]. Unlike other cell adhesion receptors, integrins require prior conformational activation by extracellular soluble mediators to bind their ligands [1]. Selectins are a family of three trans-membrane calcium-dependent lectins (L-, E-, and P-selectin) mediating cell-to-cell adhesion [2,3,4]. While L-selectin is expressed by most leukocytes, P-selectin is displayed by megakaryocytes, platelets, inflamed endothelial cells, and a subset of bone marrow (BM) endothelial cells, whereas E-selectin is expressed by endothelial cells during inflammation, as well as in specialized domains of the BM endothelium [5]. Cadherins, e.g., E-, VE- and N-cadherin, are glycoproteins that take part in cell-to-cell adhesion through the generation of intercellular junctions, providing isolation of different compartments [2,3]. Cadherins also participate in signal transduction pathways, due to their cytoskeleton anchorage [2,3]. IgSF proteins are characterized by the presence of one or more immunoglobulin-like domain(s) [2]. Most IgSF members are trans-membrane glycoproteins composed of an extracellular domain, a single trans-membrane domain and a cytoplasmic tail [6]. They mediate calcium-independent adhesion through their N-terminal domain and commonly bind other Ig-like domains on an opposing cell surface, but they can also interact with other AMs (like integrins) and carbohydrates [7]. The C-terminal domain binds to the cytoskeleton [7]. Multiprotein complexes made up of these and other AMs form cell-to-cell bindings such as anchoring, tight, and gap junctions.

Deregulation of adhesion is considered a hallmark of metastatic solid tumors that seemed to be less of an issue in acute myeloid leukaemia (AML). AML is a genetically-heterogeneous group of multi-cause malignancies [8] in which clonal, undifferentiated or aberrantly-differentiated haematopoietic cells, known as blasts, accumulate in the bone marrow, peripheral blood, and other organs. Currently, 35% to 40% of adult AML patients who are 60 years of age or younger and only five to 15% of those who are >60 years of age can be cured [9]. Despite being viewed as a “liquid” tumor, AML blasts, are not “unattached”. On the contrary, they require a close relationship with the BM microenvironment for their survival and disease progression. Worse still, mutations in BM stromal cells may lead to the insurgence of AML pinpointing just how important and “intimate” this attachment is [10,11]. AML is maintained by a pool of self-renewing cells denominated leukaemic stem cells (LSCs) [12] that are the malignant counterpart of the haematopoietic stem cells (HSCs) in normal BM. LSCs are cells capable of initiating the disease when transplanted into immunodeficient animals and also of partial differentiation into AML blasts that constitute disease bulk, but are unable to self-renew [13].

Not all of the known AMs are discussed here. Instead, we zoom in on the molecules relevant to AML, as attested to by the available literature. Thus, we review the deregulation of adhesion in AML, focusing on the altered expression pattern and signaling from AMs, the functional consequences, prognosis and the possibilities of devising targeted therapies against aberrant interactions that may help to increase the cure rates for AML patients.

## 2. Expression of Adhesion Molecules on Haematopoietic/Leukaemic Stem Cells

HSCs express a host of AMs, as detailed in Figure 1 and Table 1. The integrin class expressed on haematopoietic cells is represented by CD29 (the β1 integrin chain) dimerized with CD49a-f (α1–6 chains) to form the very late antigens (VLA-1 to 6), of which VLA-4 and VLA-5 are the most widely displayed upon HSCs and progenitors [14]. HSCs also exhibit CD18 (β2 integrin chain) dimerized with CD11a-c (αL, αM, αX chains) and forming lymphocyte function-associated antigen-1 (LFA-1), macrophage-1 antigen, or p150/95, respectively [14], as well as CD61 (β3 integrin chain) bound to CD41 (αIIb) or CD51 (αV) accordingly forming gpIIb/IIIa or vitronectin receptor [15,16]. Of the selectin class, instead, the L-selectin is present on leukocytes and immature progenitors [17]. Both E- and N-cadherins are exhibited on stromal cells and on a fraction of CD34+ progenitor cells [18,19]. Within the IgSF of importance are the vascular cell adhesion molecule-1 (VCAM-1, CD106) and intercellular adhesion molecule-1 (ICAM-1, CD54) present upon the stromal cells and interacting with VLA-4 expressed on HSCs [20].

Other types of AMs expressed by HSCs include sialomucins such as CD34, CD45R, CD43, CD162 (also known as the P-selectin glycoprotein ligand 1), and CD164 [21], as well as the CD44 [22] and syndecan-1 [23] proteoglycans (Figure 1, Table 1). CD44 is expressed by haematopoietic progenitors, although, it is present at a higher level on stromal cells, whereas a sialofucosylated glycoform of CD44, known as haemopoietic cell E-/L-selectin ligand (HCELL) is displayed exclusively upon haematopoietic cells [24]. Haematopoietic cells also exhibit the CXCR4 chemokine receptor [25], the ligand of which is the stroma-derived factor-1 (SDF-1 also known as CXCL12), necessary for colonization during development [26] and for homing during BM transplantation. Furthermore, HSCs and progenitor cells possess, on the surface, an array of ephrin receptors and ligands [27]. Of relevance is also the expression of the connexin (CX) family of gap junction proteins including CX32 (also known as gap junction protein 1β) [28] and CX43 (gap junction protein 1α) [29].

The expression pattern of AMs present on normal HSCs and AML blasts, including LSCs, is largely similar. The differential expression includes the presence of VLA-1, -2, -3, and 6 that are usually absent on normal haematopoietic cells [30]. LSCs also show the surface expression of the CD98 trans-membrane protein, a molecule that amplifies adhesive signals deriving from ECM through interactions with integrins. CD98 abrogation impairs the establishment and propagation of AML in mouse models of the disease [31]. AML blasts express the platelet/endothelial cell adhesion molecule-1 (CD31) and CD38, responsible for interactions with microenvironmental elements, the first one with BM endothelial cells, while the latter with ECM-associated hyaluronate. Excess of CD31, relative to CD38, leads to a higher rate of trans-endothelial migration. Conversely, excess of CD38 results in the arrest of AML blasts within the BM niche due to hyaluronate adhesion [32]. Additionally, connexins 25, 40, and 31.9 levels are increased in AML cell lines [33].

## 3. Normal and Leukaemic Niche

HSCs are not randomly spaced within the BM microenvironment, but are positioned in and interact with discrete units referred to as niches, which determine the fate of HSC. The niche, i.e., the coupling of haemopoietic, stromal cells, and ECM, is a functional element that permits HSCs to expand or adopt a quiescent phenotype protecting their integrity and properties [34]. There are two types of haematopoietic niches: endosteal and vascular. The main factors that fasten HSCs and progenitors to the niche, possibly inducing quiescence of HSCs are VCAM-1 and CD44 [35]. In addition, haematopoietic growth factors such as stem cell factors and FLT3 ligand, SDF-1 chemokine, growth-regulated protein β and interleukin 8 (IL-8), proteases, peptides, and other chemical transmitters such as nucleotides regulate the attachment and quiescence of HSCs [35]. In the endosteal niche a pivotal role in anchoring HSCs is exerted by β-catenin and N-cadherin, the latter is required also to regulate quiescence of HSCs and to keep them in an undifferentiated state [36]. The significance of VLA-4/VCAM-1, VLA-5/fibronectin and CD44/hyaluronan/osteopontin interactions between HSC/progenitors and stromal cells for their retention in the niche was revealed by blocking the function of VLA-4, VLA-5, and CD44 using appropriate antibodies [37,38]. These interactions determine the expansion of the quiescence-promoting microenvironment and confer resistance to chemotherapy [39,40].

Adhesion to the niche is critical to AML pathogenesis and progression. LSCs require a disease-promoting BM microenvironment that plays a role in disease initiation as it may transform HSCs [11]. Several AMs are relevant to AML [41]. VLA-4, upon interaction with VCAM-1, activates pro-survival and proliferative pathways in both AML and stromal cells via the nuclear factor kappa B (NF-κB) pathway, leading to chemotherapy resistance due to protection from apoptosis [42,43] as it allows for complete integration into the vascular niche and confers a quiescent phenotype to AML cells [44]. Similarly, upon interaction with fibronectin and stromal cells AML cells are protected from apoptosis. Particularly, the VLA-4-fibronectin interaction is decisive for minimal residual disease (MRD) in AML and subsequent relapse [43]. The second axis of cell adhesion used by AML cells is the interaction of CD44 and E-selectin [45].

Other than adhesion, AML cells are also regulated by soluble factors secreted by niche cells, such as SDF-1 or the CC ligand 2 (CCL2) chemokine. In normal haematopoiesis, the CXCR4-SDF1 system regulates leukocyte trafficking and homing [46]. Chemotherapy induces expression of CXCR4 in AML cells, leading to therapy resistance and AML blast survival. Indeed, an interaction between CXCR4 on leukaemic cells and niche-derived SDF-1, together with the VLA-4/VCAM-1 interaction, is necessary for proper homing and growth of malignant CD34+ cells [3,47,48,49]. Instead, the CCL2/CCR2 axis is expressed in the majority of monocytoid AML blasts and is involved in cell trafficking and proliferation [50].

LSCs are capable of remodeling the niche to a self-reinforcing leukaemic niche [51,52]. Niche alterations occur by means of exosomes [53], microvesicles [54], and DNA fragments [55]. Exosomes secreted by AML cells alter mesenchymal stromal cells [53]. CXCR4-expressing microparticles, which modulate BM homing of leukaemic cells, were detected in plasma samples of newly diagnosed adult AML patients and correlated with white blood count [54]. AML cell-released fragmented DNA has a drastic effect on neighboring cells as it damages the microenvironment and allows AML blasts to leave the BM and enter the bloodstream [55].

Moreover, positive feedback loops mediated by disease can cause inflammatory overexpression of AMs on activated endothelial cells [3]. AML cells express HCELL and cutaneous lymphocyte antigen-1 (CLA-1), the principal ligands of E-selectin, which may be over-expressed on endothelia due to tumor necrosis factor α production by AML cells [12,56,57]. In addition, the LFA-1 integrin, upon binding ICAM-1 (present on AML blasts) stimulates platelet derived growth factor (PDGF), endothelial growth factor (EGF), and vascular endothelial growth factor (VEGF) receptors [3,12,47,58]. Together, such stimuli cause adhesion of AML cells to the niche leading to protection from apoptosis, LSC quiescence, escape from the immune system and chemoresistance that may be followed by detachment of AML blasts and subsequent proliferation, leading to relapse [3,12,47]. The acquisition of resistance through enhanced adhesion is so relevant that the term “environment-mediated drug resistance” has been coined [59].

## 4. The Balance between Homing and Migration in Normal and Leukaemic Cells

Whereas transplanted HSCs and progenitors injected into the bloodstream home to sites of haematopoiesis, they can be mobilized from these niches into the blood through either physiologically or pharmacologically-induced mechanisms (Figure 2A). Sialomucins, selectins, and integrins play important roles in migration and homing of HSCs [4,60]. Migration implies that HSCs translocate from the endosteal to the vascular niche and subsequently enter blood vessels through transendothelial migration, and ultimately, circulate in the blood stream. When in the vessel lumen, HSCs remain in constant contact with the endothelium. HSC homing is the opposite of this process, i.e., HSCs leave the blood stream through the endothelium, reach the vascular niche and then return to the endosteal niche.

Cytokines such as stem cell factor, chemokines such as SDF-1 and IL-8, and proteolytic enzymes such as the metalloprotease superfamily are also involved in migration and homing. HCELL is of particular relevance in these processes: It is the most potent ligand for E and L-selectins that regulate cell movement (cell rolling) and promote weak HSC adhesion to BM vessels [61]. The expression of the CXCR4 chemokine receptor on HSC surface promotes cell activation via SDF-1 ligand binding. Following higher affinity interactions between LFA-1/ICAM-1 and VLA-4/VCAM-1, HSCs come to a stop on the endothelial surface and migrate across the basal lamina. Additionally, migration is promoted by VLA-4 and VLA-5 interaction with fibronectin, present in the ECM [62].

Similarly to normal HSCs, LSCs home to the BM and they exit it (Figure 2B). LSC homing is a property used for passaging murine AML cells in mice in order to expand primary leukaemic cells and create cohorts of nearly identical mice for experiments such as in vivo drug testing. Multiple signaling pathways underlie the mobilization of AML blasts. An example is the VEGFR-1, β1 integrin, and human eag-related gene-1 K+ channel complex that mediates cell migration and invasion and hence confers a higher invasiveness to leukaemic cells in vivo [63]. Moreover, the activation of the complement cascade in leukaemia/lymphoma patients (e.g., due to accompanying infections) enhances the motility of malignant cells and contributes to their spreading in a p38 mitogen-activated protein kinase–heme oxygenase-1-dependent manner [64].

Dislodging AML blasts from the niche is a way of increasing their chemosensitivity and improving the efficacy of anti-AML therapies and will be discussed in more detail in the section of targeted therapies.

## 5. Involvement of AMs in AML Signaling

In addition to their structural roles in anchorage, AMs transmit signals into the cells in response to the adhesive state, cell’s location, and surrounding ECM. Such signaling is involved in tumor development and progression. Thus, accumulating evidence indicates that integrins are implicated in leukaemia signaling. The binding of the integrin tail to adaptor proteins such as SRC kinase, focal adhesion kinase (FAK), or integrin-linked kinase (ILK) leads to the activation of different signal transduction pathways including mitogen activated protein kinase cascades, phosphoinositide-3-kinase (PI3K)/AKT kinase and protein kinase C [65,66]. ILK is constitutively active in AML [66] and favors cell growth and survival by the activation of AKT and inhibition of glycogen synthase kinase-3-β (GSK3B), therefore stabilizing β-catenin [67,68], the transcriptional co-activator in the canonical Wnt pathway [69]. MAC-1 binds spleen tyrosine kinase (SYK) and activates signal transducer and activator of transcription 3/5 in AML cells, ultimately causing cell survival and proliferation [70]. In addition, β3-integrins were found to be essential for the development of leukaemia in transgenic and xenograft mouse models [71]. Curiously, their expression is regulated by HomeoboxA9 and HomeoboxA10 transcription factors implicated in AML. Notably, the downstream signaling from β3-integrins, also causes the induction of SYK kinase and affects cell homing, transcription regulation, and induction of differentiation of LSC [72]. Furthermore, Yi et al. recently reported that αvβ3 integrin enhances β-catenin expression through the PI3K/AKT/GSK3B pathway [73].

Little is known about selectins’ signaling in AML, albeit Levesque and colleagues demonstrated that VE-selectin confers protection from chemotherapy by activation of pro-survival NF-κB signaling. Interestingly, this activation was not observed upon adhesion to P-selectin, CD31 or VCAM-1 [45]. Moreover, upon E-selectin ligand-1 or CD44 binding to E-selectin, Wnt signaling is activated and promotes AML proliferation [74].

Cadherins’ cytoplasmic domain directly binds β-catenin, which in turn is attached to α-catenin that serves as a link to actin cytoskeleton. Some evidence has suggested that cadherins can influence Wnt signaling pathway by competing for the pool of β-catenin [75]. Moreover, the phosphorylation of β-catenin through AKT, c-SRC or c-jun N-terminal kinase 2 promotes the dissociation of β-catenin from the junctions [76] decreasing cell adhesion, and inducing the transcription of target genes such as matrix metalloproteinase that degrade the ECM components influencing cell adhesion and rendering tumor cells more invasive [77].

## 6. EMT-Like Programme Activation in AML

The t(8;21)(q21;q22) translocation, which gives rise to the AML1/ETO oncogenic fusion, is among the most common rearrangements found in AML. We recently reported that AML1/ETO expression reduces adhesion and enhances migration of haemopoietic cells in vitro [78]. Such a phenotype translated into a homing defect of AML1/ETO-bearing cells in transplantation experiments in vivo [78]. The altered balance between adhesion and migration brings to mind the phenomenon of epithelial-mesenchymal transition (EMT) observed in solid tumors [79]. In recent years, research into the contribution of EMT to cancer has shed light on the mechanisms that allow a primary cancer to become invasive and metastasize. Current discoveries suggest that EMT is not merely a mechanism used by cancer cells to acquire motile phenotype, but is associated with the insurgence of stem cells able to indefinitely maintain the cancer [80]. Until recently, EMT has not been described or sought after in haematological malignancies, although there is an ever-increasing body of evidence that, indeed, an EMT-like process exists in this context. In the case of AML, tumor cells are present in the haematopoietic tissues throughout the organism right from the clinical onset of the disease. It is thought that leukaemias derive from the transformation of a single cell and its progeny that evolves in clones of various fitness. The mechanisms that underlie the spreading of preleukaemic and leukaemic cells from the primary localization to the entire haematopoietic system are unknown (Figure 2B). It stands to reason that EMT-like phenomena may play a role at these early stages of leukaemogenesis and the EMT machinery may be deployed to achieve a more mobile and less adhesive phenotype in cells bearing leukaemic oncogenes. Such changes could alter the interaction with the haematopoietic niche [81] and lead to the acquisition of features that allow immature cells to migrate across the BM barrier. In agreement with such a scenario, EMT inducers such as Twist, Zeb1, Zeb2, and Snail/Slug have been shown to play critical roles in HSCs and in leukaemia [82]. In addition, the expression of several EMT-related genes is significantly associated with poor overall survival (OS) of AML patients [83]. In particular, Zeb2 is essential for embryonic HSCs and progenitor cells differentiation in the fetal liver [84]. Zeb2 also regulates HSC numbers and the differentiation of myeloid progenitors and B-cell precursors in a mouse model bearing conditional deletion of Zeb2 in adult haematopoietic cells. Furthermore, although perhaps counterintuitive for the hypothesis of the contribution of an EMT-like process to leukaemogenesis, mice with a conditional Zeb2 deletion develop splenomegaly and a parallel increase in extramedullary haematopoiesis [85]. The knockdown of Zeb1 in MLL/AF9-driven leukaemia drastically reduces leukaemic blast invasion [83]. Experiments of retroviral insertional mutagenesis identified Zeb2 activation as a leukaemogenesis-collaborating event in CALM-AF10 transgenic mice [86]. Moreover, Zeb2 depletion impaired proliferation of human and murine AML cells and caused aberrant differentiation of human AML cells through transcriptional repression of myeloid differentiation and deregulation of the cell adhesion and migration signature [87]. Last but not least, the expression of E-cadherin (CDH1), the loss of which is considered the hallmark of EMT in solid tumors [79], is low in leukaemia due to the hypermethylation of its promoter [88].

In such a context, EMT could be responsible for (or contribute to) not only the motility, but also for the differentiation block and stemness induction functions of AML-associated oncogenes. If such a hypothesis were true, EMT could be seen as a universal step needed for every cancer to develop, and thus a common hub to be targeted for cancer therapy.

## 7. Clinical Implications

### 7.1. Prognosis

The serum levels of chemokine or adhesion receptors represent prognostic factors for AML patients with a clear impact on OS and relapse. In particular, expression levels of VLA-4 and CXCR4 have been associated with patients’ outcome. Several independent studies showed that a high level of expression of CXCR4 predicts low rates of OS and event-free survival, while a low level correlates with increased OS, relapse free survival and complete remission (CR) rate [89,90,91,92]. On the other hand, elevated expression of VLA-4 has been correlated with longer survival for paediatric patients affected by AML [93], whilst increased binding of soluble VCAM-1 via VLA-4 was significantly associated with longer OS corrected for age in untreated (de novo and secondary) adult AML patients [94]. A recent study measured the expression of VLA-4 and CXCR4 in BM aspirates of 98 patients with newly diagnosed AML, and proved that the level of VLA-4 was higher in patients that belong to favorable and intermediate risk classes. Moreover, subjects with high expression of VLA-4 had more probability to achieve CR and a lower risk of relapse. Contrarily, CXCR4 expression did not correlate with a different cytogenetic risk category, although high expression of this chemokine increased the risk of relapse. Thus, patients with low CXCR4 and high VLA-4 expression levels were characterized by a better outcome in terms of OS and relapse-free survival in comparison to those with high CXCR4 and low VLA-4-expression levels [95]. Phosphorylation of the Serine339 residue of CXCR4, which impairs the mobilization induced by CXCR4 inhibitors, was associated with poor prognosis in AML patients and has been implicated in resistance to cytarabine treatment [96].

High level of integrin β3 expression is associated with shorter OS of AML patients, especially in cases with FLT3-ITD mutations. The expression of integrin β3 was higher in the poor risk group than in the favorable and intermediate groups [73].

A prognostic value of other AMs, e.g., FAK and protein tyrosine kinase-2 (PYK2), has been assessed [97]. AML patients treated with intensive regimens were characterized by a heterogeneous expression of FAK and PYK2, both of which did not correlate either with clinical or cytogenetic features. The OS was significantly longer for patients with a lower expression of FAK, but did not correlate with PYK2 levels. Furthermore, another study appraised the prognostic value for OS prediction focusing on the expression of three different markers: CXCR4, VLA-4, and FAK [92]. Subjects that had an overexpression of one out of three markers had a longer OS than patients overexpressing two or three factors [92]. The prognostic utility of CD44 was also analyzed. Expression of CD44 turned out to be useful as a prognostic marker for elderly AML patients in whom high expression of CD44 is associated with a reduction of OS [98]. While not predictive for the outcome, high expression of syndecan-1 was found to have a clinical relevance as it is associated with bleeding thrombocytopathy, endothelial cell damage, and leukocytosis [99].

Interestingly, important EMT markers seem to have prognostic relevance. In particular, a poor clinical outcome is linked to elevated mRNA expression of vimentin. Analysis of mRNA expression data from 173 AML patients from the cancer genome atlas dataset, suggested that patients older than 60 years, with a normal karyotype and high vimentin expression have poor clinical outcome [100]. Similarly, low levels of CDH1 expression were found to be of prognostic value in normal karyotype AML, correlating with a markedly shorter OS [101].

### 7.2. Targeted Therapies

Currently, no therapeutic strategies for AML patients perform better than conventional chemotherapy. Sadly, the outcome remains poor and the standard of cure represents an option that is not always applicable as many patients are unfit for intensive regimes [102,103]. Thus, new alternatives need to be exploited and the interaction between AML cells and the BM microenvironment may represent a valid target (Figure 3). However, such a strategy remains a challenge as it may also eliminate normal HSCs [4,47,104,105,106]. Several preclinical and early phase clinical trials using agents that target the AML-niche interaction confirmed the influence of the microenvironment on proliferation, differentiation, and apoptosis of AML blasts. The possible targetable AML-stroma interactions include AMs, CXCR4/SDF-1 signaling, and hypoxia.

The AM-ligand interactions harnessed in AML therapy so far include: VLA-4 with VCAM-1, VLA-4 and CD44 and E-selectin, E-selectin ligand-1 and E-selectin, as well as the integrin/CD44 interaction with osteopontin, a glycoprotein of the ECM.

AS101 is an agent against VLA-4 targeting the fibronectin-bound form. Based upon in vivo studies, AS101 inhibits the PI3K/Akt signaling pathway and acts as chemosensitizer of chemoresistant cells [107]. An ongoing clinical trial is investigating the efficacy of AS101 in combination with chemotherapy, for elderly patients affected by AML and myelodysplastic syndrome [108]. Moreover, FNIII14, a VLA-4 antagonist, helps to overcome the drug resistance mediated by cell adhesion and, when administered in concert with standard treatments, successfully eradicated MRD [109]. Encouraging in vivo results were similarly achieved with a FDA-approved humanized anti-VLA-4 antibody, Natalizumab. However, its utility has been limited due to the unforeseen progressive multifocal leukoencephalopathy [110]. The in vitro experiments of blocking αvβ3 integrin with antibodies that enhanced AML cell lines sensitivity to sorafenib [73] and of knockdown of CX25 that sensitized AML cell lines to chemotherapeutic agents are also promising [33].

Inhibition of E-selectin employing a specific small molecule, GMI-1271, augments the effect of chemotherapeutic agents and decreases tumor burden in xenograft mouse models [74]. A current phase I clinical study has demonstrated the safety and the pharmacokinetics of this compound on healthy subjects, whilst the evaluation of its activity on haematological malignancies is ongoing [111]. Lastly, it has been shown that inhibition of the osteopontin pathway induces the exit from quiescence of LSCs, reduces homing, and increases sensitivity to cytarabine treatment in engrafted mice [39].

Given CXCR4/CXCL12 involvement in homing, quiescence and proliferation of AML blasts [112] it has long been explored as a target. CXCR4 inhibitors, usually combined with standard treatment, have given positive results both in vitro and in vivo, as well as in AML clinical trials (phase I/II). Preclinical studies reported that CXCR4 inhibitors decrease adhesion and migration through stromal and endothelial cell monolayers, induce cell differentiation, and abrogate the protective effect of BM stromal cells thereby enhancing cell apoptosis and chemosensitivity [49,113,114,115,116,117,118]. AMD3100, also known as plerixafor, is an FDA-approved CXCR4 inhibitor. An increase in blast mobilization, a CR, and CR with incomplete haematological recovery (CRi) accompanied by few adverse effects were observed in patients with both newly diagnosed and relapse/refractory AML treated with AMD3100 in addition to chemotherapy [119,120,121]. Other clinical trials underlined a positive effect of Ulocuplumab [122,123], BL-8040 (BLT40) [124,125] and LY2510924 [126], a human IgG4 monoclonal antibody against CXCR4 and two peptidic CXCR4 antagonists, respectively, in terms of the mobilization of blasts, induction of apoptosis, differentiation of leukaemia cells, and achievement of CR and a CRi in patients with relapsed/refractory AML. Finally, a polysaccharide derived from heparinoids, known as CX-01, was shown to disrupt the CXCR4/CXCR12 axis and induce, together with chemotherapy, a morphological remission in AML patients [127,128].

An alternative approach for targeting the CXCR4/CXCR12 axis is via the inhibition of CXCR12. Recently, a study revealed that NOX-A12, a CXCR12 inhibitor, induces chemosensitisation and interferes with migration of chronic lymphocytic leukaemia (CLL) cells [129]. It is being currently tested in phase II trials in relapsed CLL and multiple myeloma patients [130]. While all these agents are well tolerated, additional clinical trials (phase II/III) are needed to optimize the combination and avoid the major adverse effect of CXCR4 inhibitors such as hyperleukocytosis [131].

An additional factor that impairs the effect of chemotherapy is the hypoxic BM microenvironment and the resulting hypoxia-inducible factor-1α transcription factor induction which in turn upregulates the expression of AMs (selectin ligands, syndecan-4 or α5 integrin [132]) or CXCR4 [133] on cancer cells. AML cells localized in the hypoxic niche are exposed to lower amounts of chemotherapeutic agents [134]. Furthermore, hypoxia boosts angiogenesis and cytokine secretion, both of which are involved in resistance to chemotherapy [133]. Indeed, AML cells derived from chemo-resistant patients treated with TH-302, a prodrug that under hypoxic conditions releases the DNA cross-linker bromo-isophosphoramide mustard, became sensitive to standard chemotherapy [135]. A phase I study demonstrated a transient cytoreduction in all refractory AML patients [136].

Other strategies aimed at increasing mobilization and response to drug treatment are under investigation. Indeed, targeting the kinases involved in the phosphorylation of CXCR4 (e.g., PIM1, GRK6, or PKC) with small-molecule inhibitors may decrease the retention of AML cells in the BM niche [96].

## 8. Conclusions

Adhesion plays an important role in the physiology and pathology of tissue homeostasis. Several of the findings in the adhesion field are surprising and may be non-instinctive: AML is more “solid” than it seems, adhesion loss is not equal to cancer spreading and EMT plays a role in the progression of a non-epithelial tumor. For years, the notion that loss of adhesion leads to cancer metastasis has been considered a paradigm. An ever-growing body of evidence calls for paradigm shift by suggesting that the opposite may be true [137]. Indeed, adhesion molecules are not only responsible for the joining of cells, but are also receptors whose activation leads to downstream intracellular signaling that contributes to (or drives) tumor progression. Accordingly, as shown in this review, the interactions between AML blasts and the BM niche influence haematopoiesis, leukaemogenesis, cell survival, and chemotherapy response. When reviewing the literature, we noticed a circa 15-year-gap in AML adhesion research. There has been a wave of surface molecule studies in the 1990s that came to an end at the beginning of the 21st century followed by a more functional and mechanistic phase in recent years. It is hard to draw conclusions and make statements as both protein names and AML classification system have considerably changed since. Moreover, AM expression was studied in diverse patient cohorts, using a variety of methods and looking at different cellular populations (e.g., bulk in some studies versus CD34+ cells in others).

Current research explores new aspects, such as the mechanism by which physical forces and local pliability regulate adhesion and it seems that the signals they produce are of particular interest in tumor biology [138]. The role of EMT-like process in AML merits additional dissection. Furthermore, the contribution of AMs to symmetric/asymmetric LSC division and the role of ECM in leukaemia warrant more work.

Overall, much progress has been made; however, some questions still await an answer and only a small number of therapeutic agents have emerged. It is certain that there is a great potential for devising novel adhesion-related diagnostic, prognostic, and therapeutic tools. For example, syndecan-1 that is expressed on AML cells, is a therapeutic target under investigation in multiple myeloma [139], while the inhibition of ILK has been tested in chronic lymphocytic leukaemia, prostate, and breast cancer cell lines [140]. There are also some suggestions that therapy targeting the LSC-niche interaction should be sequential to debulking and serve the precise aim of LSC eradication. Taken together, the full comprehension of the mechanisms that lay beneath the adhesive interactions and signaling will help to identify rational adhesion-targeting treatments to improve the curability of AML.

## Figures and Tables

**Figure 1 cells-08-00066-f001:**
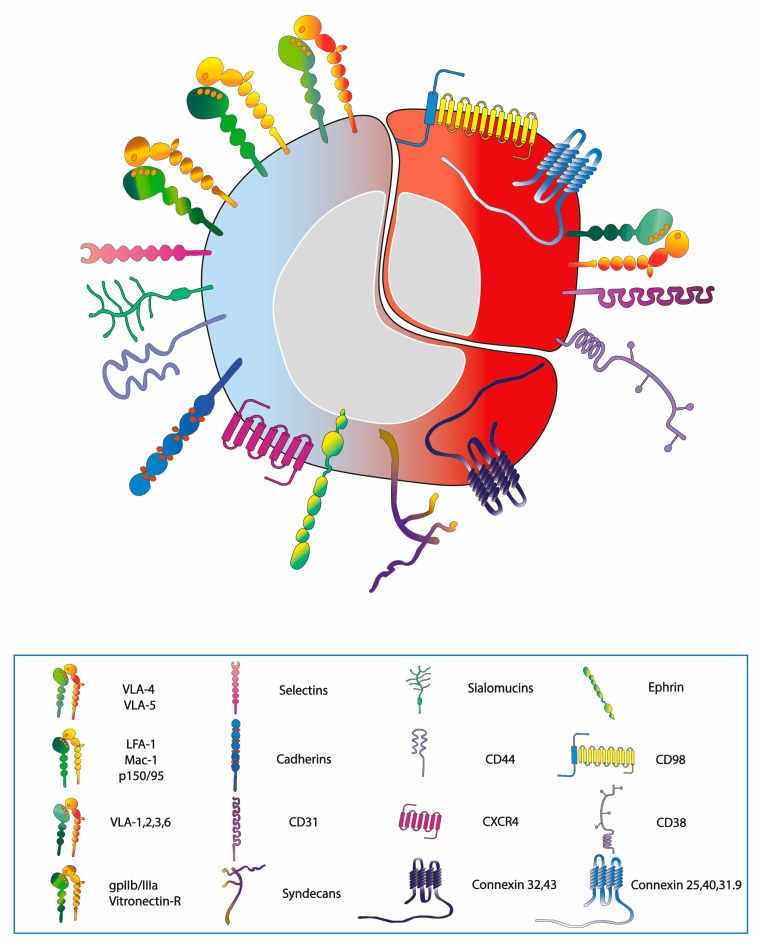
Expression pattern of AMs, chemokine receptors and other adhesion-modulating proteins on the surface of HSC and LSC. The molecules in common are depicted on the blue-red cell (left to lower right), while the molecules expressed exclusively on LSC are drawn on the surface of the red cell (upper right).

**Figure 2 cells-08-00066-f002:**
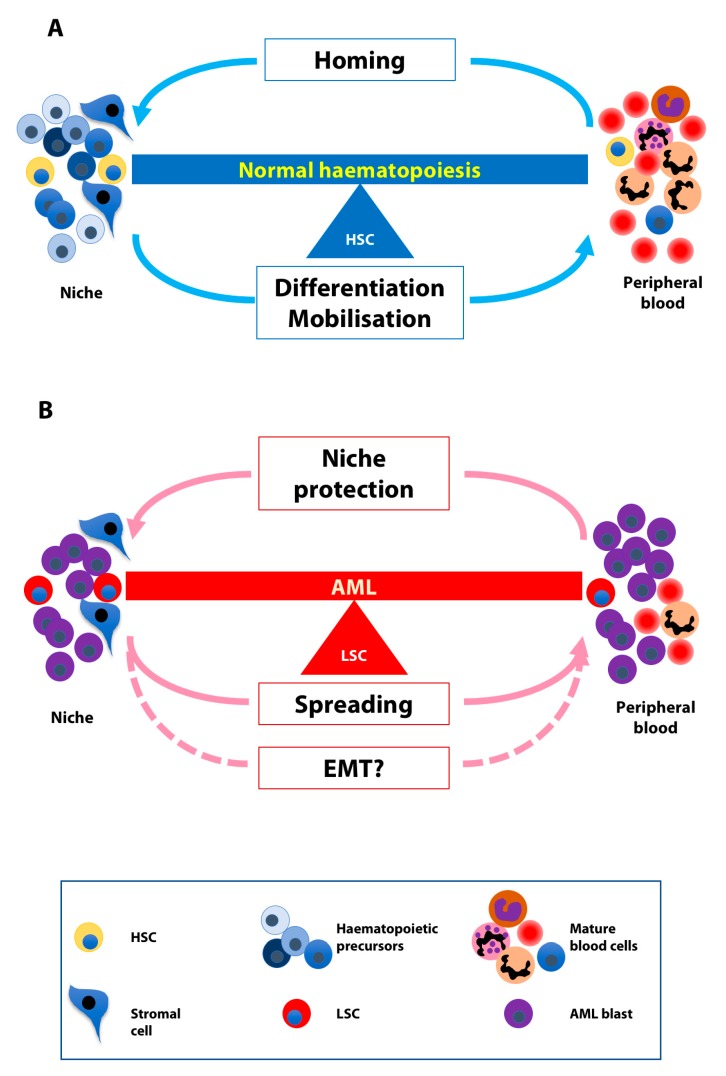
The balance between homing and migration in normal and leukaemic cells. (**A**) Normal HSCs home to the bone marrow niches and exit the niches in response to differentiating and mobilizing stimuli. (**B**) LSCs use the niche for protection from chemotherapeutic agents and detach from it in order to spread possibly deploying the EMT machinery.

**Figure 3 cells-08-00066-f003:**
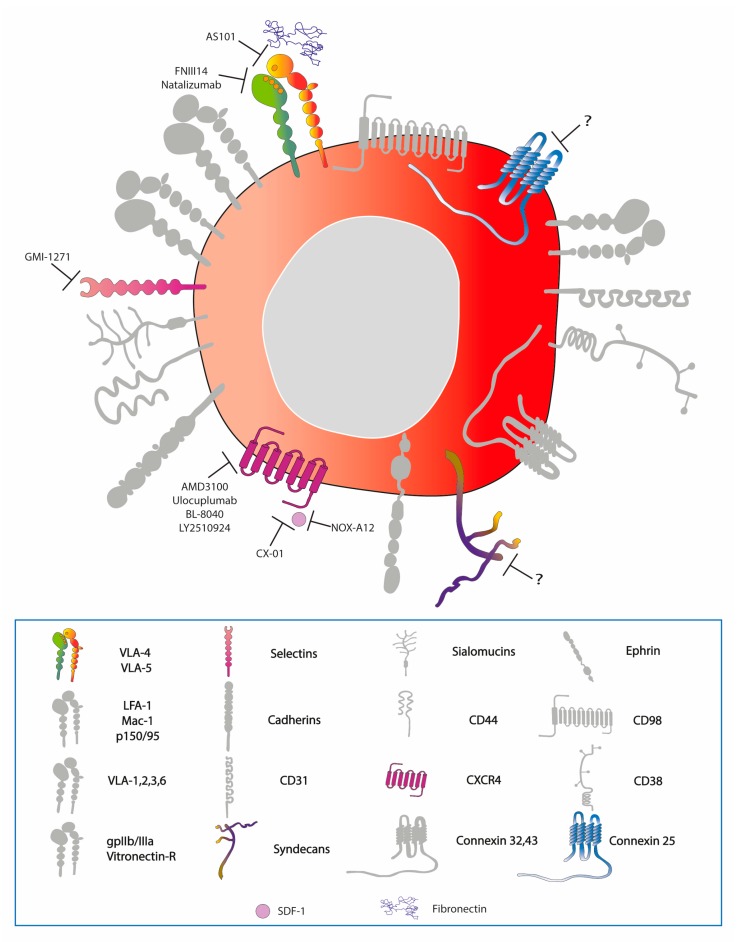
Therapeutically targetable AMs and interactions between AMs and their ligands on the surface of LSCs. The question mark denotes inhibitors of AMs targeted in other malignancies or in vitro.

**Table 1 cells-08-00066-t001:** Summary of the adhesion molecules discussed in this review. All abbreviations as detailed in the text, C3b, complement 3b, and FGF, fibroblast growth factor.

Classification of Cell Adhesion Molecules and Other Adhesion-Modulating Proteins Relevant to AML			
Family	Adhesion Molecule	Distribution	Extracellular Ligands
Integrin	VLA-1, VLA-2	LSC	Collagen, Laminin
	VLA-3	LSC	Collagen, Laminin, Fibronectin
	VLA-4	HSC/Progenitors/LSC	Fibronectin, VCAM-1, ICAM-2
	VLA-5	HSC/Progenitors/LSC	Fibronectin, Invasin
	VLA-6	LSC	Laminin, Merosin, Kalinin, Invasin
	LFA-1	HSC/LSC	ICAM-1, ICAM-2, ICAM-3
	MAC-1		C3b, ICAM-1, Factor X, Fibrinogen
	p150/95		C3b, Fibrinogen
	gpIIb/IIIa Vitronectin-R	HSC/LSC	Fibronectin, Fibrinogen, von Willenbrand factor, Vitronectin
Selectin	L-selectin	HSC/LSC	ICAM-1, Sialomucins
	E-selectin	Stromal cells	Sialomucins, CLA-1
	P-selectin	HSC/LSC/Stromal cells	Mucin-like molecules
IgSF	VCAM-1	Stromal cells	VLA-4
	ICAM-1	Stromal cells	LFA-1, MAC-1
	CD31	HSC/LSC/Stromal cells	Vitronectin-R
Cadherin	E,N-cadherin	CD34+ progenitors/ Stromal cells	Other cadherins
	VE-cadherin	Stromal cells	
Sialomucin	CD34, CD45R, CD43, CD162, CD164	HSC/LSC	Selectins
Other adhesion molecules	CD44, HCELL	HSC/LSC/Stromal cells	Hyaluronan, Osteopontin
	Syndecans		Integrins, FGFs, VEGFs, PDGFs
	Connexins		Connexins
Adhesion-modulating proteins	Ephrin receptors	HSC/Progenitors	Ephrins
	CD98	HSC/LSC	Integrins
	CD38	HSC/LSC	CD31
Chemokine receptors	CXCR4	HSC/LSC	SDF-1
Signal transducers	FAK, PYK2, ILK	HSC/LSC	Integrins

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
