# Peer review of "Adhesion Deregulation in Acute Myeloid Leukaemia"

_cells, 2019, doi:10.3390/cells8010066_

Round 1
Reviewer 1 Report
Overall this review article is well written and will be of use to the field. This reviewer only has minor comments that need to be addressed. They are as follows;
The presentation of the adhesion molecules presented gives the impression that these "main" molecules highlighted are the only ones that exist that are relevant to AML. This is surprising as there are a host of other AM that have been linked to AML and could be potential therapeutic targets as well that were not discussed here. Syndecans and gap junctions for example have been suggested in recent literature. If the authors truly want this to be a comprehensive review its worth mentioning these other AMs.
The conclusion was found to be very brief and pretty much restated what was in the abstract. Adding a bit more critical analysis of the proposed targets and how this could cause a paradigm shift in the field, for example would be helpful here
Author Response
Reviewer 1
Overall this review article is well written and will be of use to the field. This reviewer only has minor comments that need to be addressed. They are as follows;
We thank the reviewer for taking time to review and comment on the manuscript.
The presentation of the adhesion molecules presented gives the impression that these "main" molecules highlighted are the only ones that exist that are relevant to AML. This is surprising as there are a host of other AM that have been linked to AML and could be potential therapeutic targets as well that were not discussed here. Syndecans and gap junctions for example have been suggested in recent literature. If the authors truly want this to be a comprehensive review its worth mentioning these other AMs.
We thank the reviewer for pointing out the omission of some of the AMs that are linked to the pathogenesis and that may potentially be target in AML therapy. We have included the syndecans and the connexins.
The conclusion was found to be very brief and pretty much restated what was in the abstract. Adding a bit more critical analysis of the proposed targets and how this could cause a paradigm shift in the field, for example would be helpful here.
The conclusion section has been made longer in order to accommodate the comments of both reviewers. We have included the concept of a paradigm shift and examples of yet unexplored targets have been provided.
Reviewer 2 Report
Gruszka et al had put together a nice written review describing the role of adhesion and adhesion molecules (AM) in acute myeloid leukemia. They summarize the expression of AMs in hematopoietic and AML stem cells, they move to examine the role of AM in the bone marrow niche and how that affect normal hematopoiesis and leukemia. They also describe how EMT-like process contribute to AML. Finally they review the prognostic value of some AMs in AML and summarize the status of targeting AMs as therapeutic approach in AML. Overall, the review is interesting and will provide a useful summary for AMs in AML. However, there are few comments/changes that I think would help to significantly improve the manuscript:
1- I suggest the authors provide a definition of what is AMs are, and a table for the AMs that were covered in their review, and which class they belong to. They mention CD38, PAK, PYk2, what classes do these AM belong to?
2- More information about acute myeloid leukemia is needed at “Introduction” part. There is little background introduction on AML at present. The additional information includes: what is AML, what’s the role of LSCs and progenitors in AML.
3- There is significant overlap and repetition in information mentioned in the introduction and in sections 2 (expression of AM…), similarly between section 2 and 3 (Normal and leukemia niche). The introduction can be modified to provide an outline of the review and what will be discussed.
4- How adhesion molecule is associated with particular mutations in AML, if any. Or particular clinical or molecular characteristics
5- The review does not mention the role of AM in cell signaling and how they function mechanistically, I think it is important to include this part in the review, ILK signaling and downstream pathway for example plays an important role in AML.
6- There is switch back and forth between HSC and AML cells, I think separating the role of AM in hematopoiesis from that in leukemia would make the review easier to read.
7- The review should end with describing the limitation in knowledge in the field of AMs in AML, and what areas should future research focus on, in other word what are the remaining unanswered questions in the field.
Minor comments:
Line 64: change “concentration “ to “focusing”
Line 113: there should “is” after the latter.
Line 326-333: it is not clear how this part is related to adhesion, further clarification would be needed.
The order numbers of the sections “normal and leukaemic niche” and “the balance between homing and migration in normal and leukaemic cells” are repeated.
Author Response
Reviewer 2:
We have read with attention the comments and requests provided by reviewer 2. We thank the reviewer for useful suggestions that we have implemented as follows:
1- I suggest the authors provide a definition of what is AMs are, and a table for the AMs that were covered in their review, and which class they belong to. They mention CD38, PAK, PYk2, what classes do these AM belong to?
We have added a definition of the term AM and put together a table that comprises all of the molecules mentioned in the text.
2- More information about acute myeloid leukemia is needed at “Introduction” part. There is little background introduction on AML at present. The additional information includes: what is AML, what’s the role of LSCs and progenitors in AML.
The introductory paragraph has been extended to include more information on AML and LSCs.
3- There is significant overlap and repetition in information mentioned in the introduction and in sections 2 (expression of AM…), similarly between section 2 and 3 (Normal and leukemia niche). The introduction can be modified to provide an outline of the review and what will be discussed.
We have reread critically the manuscript and significantly reduced the overlap and the repetitions between the sections.
The introduction has been modified to list all of the issues included in the review.
4- How adhesion molecule is associated with particular mutations in AML, if any. Or particular clinical or molecular characteristics?
There are no published papers on the correlation between the expression of individual AMs and the incidence of AML-associated mutations/molecular characteristics. We have attempted to perform an analysis using the BloodSpot tool (Nucleic Acids Research, Volume 47, Issue D1, 8 January 2019, Pages D881–D885) in order find such correlation for five types of AML (t(15;17), inv(16), t(8;21), MLL rearrangements and complex karyotype) present in the dataset. Although, there seem to be some differences, the results are not easy to interpret and we are reluctant to make firm statements. In fact, BloodSpot takes advantage of deposited RNA gene expression profiles that may not correspond to protein levels. Therefore, more experimental work would be needed to confirm any possible associations that we are unable to perform and would be beyond the scope of this review.
The clinical relevance of AM expression is described in the Prognosis section (6.1).
5- The review does not mention the role of AM in cell signaling and how they function mechanistically, I think it is important to include this part in the review, ILK signaling and downstream pathway for example plays an important role in AML.
We thank the reviewer for this suggestion as its incorporation made the manuscript more comprehensive. A new section on signaling and non-adhesive functions of the AMs has been added.
6- There is switch back and forth between HSC and AML cells, I think separating the role of AM in hematopoiesis from that in leukemia would make the review easier to read.
We did consider to make the change suggested by the reviewer, but we feel that the layout adopted by us, is unusual and novel. Most published reviews talk of one thing or the other or when they make comparisons they create separate sections for normal and malignant haematopoiesis. We feel that an immediate comparison may be useful.
7- The review should end with describing the limitation in knowledge in the field of AMs in AML, and what areas should future research focus on, in other word what are the remaining unanswered questions in the field.
The conclusion section has been extended according to the suggestion of both reviewers. In particular, we have mentioned the concepts that require a clarification in the field.
Minor comments:
Line 64: change “concentration “ to “focusing”
Done
Line 113: there should “is” after the latter.
Done
Line 326-333: it is not clear how this part is related to adhesion, further clarification would be needed.
A clarification has been provided.
The order numbers of the sections “normal and leukaemic niche” and “the balance between homing and migration in normal and leukaemic cells” are repeated.
We have corrected the numbering.